# Targeting Insulin-Like Growth Factor 1 Receptor Delays M-Phase Progression and Synergizes with Aurora B Inhibition to Suppress Cell Proliferation

**DOI:** 10.3390/ijms21031058

**Published:** 2020-02-05

**Authors:** Akane Yamagishi, Yuki Ikeda, Masayoshi Ikeuchi, Ryuzaburo Yuki, Youhei Saito, Yuji Nakayama

**Affiliations:** Department of Biochemistry & Molecular Biology, Kyoto Pharmaceutical University, Kyoto 607-8414, Japan; ky14374@ms.kyoto-phu.ac.jp (A.Y.); ky15020@ms.kyoto-phu.ac.jp (Y.I.); kd17001@ms.kyoto-phu.ac.jp (M.I.); yuki2019@mb.kyoto-phu.ac.jp (R.Y.); ysaito@mb.kyoto-phu.ac.jp (Y.S.)

**Keywords:** IGF1R, IGF1, OSI-906, NVP-ADW742, M phase, ZM447439

## Abstract

The insulin-like growth factor 1 receptor (IGF1R) is a receptor-type tyrosine kinase that transduces signals related to cell proliferation, differentiation, and survival. IGF1R expression is often misregulated in tumor cells, but the relevance of this for cancer progression remains unclear. Here, we examined the impact of IGF1R inhibition on cell division. We found that siRNA-mediated knockdown of IGF1R from HeLa S3 cells leads to M-phase delays. Although IGF1R depletion causes partial exclusion of FoxM1 from the nucleus, quantitative real-time PCR revealed that the transcription of M-phase regulators is not affected by decreased levels of IGF1R. Moreover, a similar delay in M phase was observed following 2 h of incubation with the IGF1R inhibitors OSI-906 and NVP-ADW742. These results suggest that the M-phase delay observed in IGF1R-compromised cells is not caused by altered expression of mitotic regulators. Live-cell imaging revealed that both prolonged prometaphase and prolonged metaphase underlie the delay and this can be abrogated by the inhibition of Mps1 with AZ3146, suggesting activation of the Spindle Assembly Checkpoint when IGF1R is inhibited. Furthermore, incubation with the Aurora B inhibitor ZM447439 potentiated the IGF1R inhibitor-induced suppression of cell proliferation, opening up new possibilities for more effective cancer chemotherapy.

## 1. Introduction

The insulin-like growth factor 1 (IGF1) receptor is a receptor-type tyrosine kinase composed of two α and two β subunits, which are generated by cleavage of the IGF1 receptor (IGF1R) precursors [1]. Binding with IGF1 activates its tyrosine kinase activity and results in autophosphorylation, which can provide docking sites for insulin receptor substrate 1 (IRS1) and Shc. These proteins are phosphorylated and trigger downstream signaling pathways, such as phosphatidylinositol 3-kinase (PI3K)/AKT and Raf/mitogen-activated protein kinase (MEK)/extracellular signal-regulated kinase (ERK) pathways, which are the main intracellular effectors of IGF1/IGF1R pathways for cell proliferation, differentiation, and survival. Associations between increases in plasma IGF levels and cancer risk have been suggested [2,3,4,5]. In addition, the correlation of IGF1R expression levels with the progression of cancer and metastatic phenotypes has been reported [6,7,8,9,10]. However, several reports have shown an inverse or no correlation between IGF1R expression levels and malignancy and survival rates [5,11,12,13,14]. Further understanding of IGF1/IGF1R signaling is required to elucidate the significance of IGF1R expression levels in cancer development and malignancy.

In cell division, replicated chromosomes are aligned at the cell equator by mechanical forces provided by microtubules; through polymerization and depolymerization, microtubules carry the chromosomes to the cell equator and keep the mitotic spindle at the center of the cell by tethering it to the cell cortex [15]. To endure the pulling force from microtubules, actin remodeling is a prerequisite; cortical actin induces a round cell shape and supplies cortical rigidity [16]. These dynamic cellular events are regulated primarily by the Serine/Threonine (Ser/Thr) kinases [17]. Ser/Thr kinase cyclin-dependent kinase 1 (CDK1)/cyclin B1 phosphorylates a variety of proteins and orchestrates mitotic entry and progression. Other Ser/Thr kinases, including Aurora A, Aurora B, and PLK1, regulate mitotic progression at different places and times. Deregulation of cell division gives rise to asymmetrical chromosome separation, leading to chromosomal instability (CIN), a hallmark of cancer.

The insulin receptor (IR) reportedly regulates cell division by affecting nuclear localization of the transcription factor Forkhead box M1 (FoxM1), which transactivates a variety of genes related to M-phase progression [18,19]. Because the IR is highly homologous to IGF1R, we hypothesized that IGF1R would play a comparable role in cell division by affecting nuclear localization of FoxM1 and that IGF1R inhibition would result in abnormal cell division by decreasing transactivation activity of FoxM1. It has been reported that IGF1R signals participate in cell-cycle progression; the duration of the G2/M phase was prolonged in IGF1-null mice uterine cells [20]. These findings are consistent with the hypothesis that IGF1R regulates M-phase progression. If IGF1R regulates cell division, the deregulation of IGF1R signals could cause aberrant cell division, leading to the generation of aneuploid cells. However, whether IGF1R actually regulates cell division remained unknown.

Here, we investigated the roles of IGF1R in cell division. Treatment with IGF1R inhibitors for 2 h induced M-phase delay, accompanied by prolonged prophase/prometaphase and metaphase. In addition, the inhibitor of the spindle assembly checkpoint (SAC) AZ3146 canceled the M-phase delay caused by IGF1R inhibition. These results suggest that SAC is involved in IGF1R inhibitor-induced M-phase delay. Furthermore, the Aurora B inhibitor enhanced IGF1R inhibitor-induced suppression of cell proliferation, suggesting a potential for simultaneous use of IGF1R and Aurora B inhibitors in novel approaches to cancer chemotherapy.

## 2. Results

### 2.1. Delay in M-Phase Progression by IGF1R Knockdown

To investigate the role of IGF1R in M-phase progression, IGF1R was knocked down by siRNA treatment (Figure 1A), and M-phase progression was examined after release from the RO-3306–induced cell-cycle arrest at the G2/M border (Figure 1B–E). RO-3306 is a reversible inhibitor of CDK1; washing out of cells, therefore, makes cells resume the cell cycle, resulting in approximately 30% of cells synchronously entering the M phase and undergoing cell division [21]. After 1 h of release from RO-3306 arrest, cells were fixed and stained for α-tubulin and DNA. On the basis of these morphologies, M-phase cells were classified into four groups: prophase/prometaphase (P/PM), metaphase (M), anaphase/telophase (A/T), and cytokinesis. More than half of untransfected cells and non-targeting siRNA (siCtrl)-transfected cells began chromosome segregation, progressing to anaphase, telophase, or cytokinesis (Figure 1C, blue arrows; D, A/T, Cyto). Conversely, when IGF1R was knocked down, more than half of the cells remained before anaphase onset (Figure 1C, pink allows; D, P/PM, M), indicating that IGF1R knockdown delayed M-phase progression. We used two siRNAs targeting IGF1R to exclude off-target effects and observed M-phase delay in both siIGF1R-treated cells. The mitotic index, the ratio of the number of M-phase cells to total cell number, was not affected by IGF1R knockdown (Figure 1E). These results suggest that IGF1R plays a role in M-phase progression but not in a mitotic entry. 

To explore which sub-phase was prolonged, cells were synchronized with RO-3306, and just after release from the arrest, time-lapse imaging was performed in the presence of Hoechst 33342 to visualize DNA (Figure 2A). Although no severe morphological defects in M-phase progression were observed, it took longer for IGF1R knockdown cells to align all chromosomes to the cell equator (Figure 2A, prolonged). Some IGF1R knockdown cells showed multiple blebs with condensed chromosomes after chromosome alignment (Figure 2A, blebbing). Misoriented spindles were also observed in both control siRNA- and siIGF1R-transfected cells (Figure 2A, misoriented), suggesting that this phenotype does not depend on IGF1R knockdown. To quantitatively analyze M-phase delay in IGF1R knockdown cells, cells were classified into three groups: prophase/prometaphase (P/PM), metaphase (M), and anaphase/telophase (A/T); the duration time for each sub-phase is shown in Figure 2B. Mean duration data revealed that the duration of P/PM was extended from 23.6 to 32.1 min by IGF1R knockdown. Conversely, that of M was slightly extended, being 30.6 min in siCtrl and 34.7 min in siIGF1R, suggesting that IGF1R knockdown caused defective chromosome alignment (Figure 2B). The ratio of cells in a sub-phase is also shown in the graph, in which the peaks of these sub-phase ratios are shifted rightward upon IGF1R knockdown (Figure 2C). That is, while the peak of metaphase cells was at 30 min in the control cells (siCtrl), it was at 40 min in siIGF1R-transfected cells. Similarly, the peaks of anaphase cells were at 40 and 60 min in siCtrl- and siIGF1R-transfected cells, respectively. These results suggest that IGF1R knockdown delays chromosome alignment and anaphase onset. 

### 2.2. Effect on FoxM1-Mediated Transcription of M-Phase Regulators

One plausible explanation for this M-phase delay may be a reduction of M-phase regulators via suppression of FoxM1, as it has been reported that IR, which is highly homologous to IGF1R, stimulates the transcriptional activity of FoxM1 [18]. Because ERK, which is downstream of IGF1R signals, is known to regulate FoxM1 nuclear localization [22], FoxM1 nuclear localization was examined after IGF1 treatment. When HeLa S3 cells were serum-starved for 24 h, FoxM1 sub-cellular localization differed depending on cells (Figure 3A). Upon treatment with 0.1 µg/mL of IGF1 for 24 h, more cells showed nuclear localization of FoxM1. Quantification of FoxM1 fluorescence intensities within the nuclear area showed that IGF1 treatment increased intensities in the nuclei (Figure 3B). Western blotting (WB) revealed that 0.1 µg/mL was sufficient to trigger an IGF1/IGF1R signal, including phosphorylation of IGF1R and AKT. FoxM1 expression levels were not increased by IGF1 treatment (Figure 3C), confirming that IGF1 enhanced nuclear localization of FoxM1 but did not increase the expression level. To confirm that IGF1R also regulates FoxM1 nuclear localization, cells were transfected with siRNA targeting IGF1R (Figure 3D). When cells were cultured in the presence of fetal bovine serum (FBS), FoxM1 was localized at the nuclei in approximately 80% of cells. Control siRNA did not affect this FoxM1 nuclear localization, whereas IGF1R knockdown increased cytoplasmic localization. These results suggest that IGF1R regulates FoxM1 sub-cellular localization. However, considering that FoxM1 nuclear localization was dramatically reduced by MEK inhibitor U0126 (Figure 3E), FoxM1 nuclear localization may be only partially regulated by IGF1R.

IGF1R-induced nuclear localization of FoxM1 raised the possibility that IGF1R signals regulate M-phase progression via activation of FoxM1 transcriptional activity. To investigate this, cells were synchronized at the late G2 phase by release from thymidine treatment, as M-phase regulators accumulate during G2. In this cell-cycle synchronization, we confirmed in advance that cyclin B1 was increased at mRNA and protein levels with cell-cycle progression from the S phase (Figure 4A,B 0 h) to late G2 (Figure 4A,B, 8 h). Using this synchronization method, IGF1R knockdown cells were synchronized at late G2. Transcription of FoxM1 target genes, such as cyclin B1, CDK1, Aurora A, Aurora B, PLK1, CDC25B, CENPB, and CENPF, were examined by quantitative real-time PCR experiments. Contrary to our expectations, these mRNA levels were not affected by IGF1R knockdown (Figure 4C). Western blot analysis confirmed that expression levels of cyclin B1 and PLK1 were unaffected (Figure 4D). These results suggest that, although IGF1R partially regulates FoxM1 nuclear localization, the effect is insufficient to affect FoxM1 transcription activity. Taken together, these results suggest that IGF1R’s involvement in M-phase regulation differs from FoxM1-mediated transcription of M-phase regulators.

### 2.3. Delay in M-Phase Progression by IGF1R Inhibitors

Protein knockdown is assumed to, directly and indirectly, affect the M phase. One of the indirect effects involves the transcription of M-phase regulators. However, expressions of some M-phase regulators were not changed at the mRNA and protein levels, as depicted in Figure 4. We, therefore, investigated the direct effect of IGF1R inhibition on M-phase progression by using IGF1R inhibitors PQ401, OSI-906, and NVP-ADW742. OSI-906 is a dual IGF1R/IR kinase inhibitor, and PQ401 and NVP-ADW742 are selective IGF1R kinase inhibitors. First, to determine a half-maximal inhibitory concentration (IC50) of these inhibitors, cells were cultured for 48 h with these inhibitors at the concentrations indicated in the graph (Figure 5A). Cell numbers were then estimated using water-soluble tetrazolium (WST)-8 (Cell Counting Kit-8) by measuring the absorbance of reduced 2-(2-methoxy-4-nitrophenyl)-3-(4-nitrophenyl)-5-(2,4-disulfophenyl)-2H-tetrazolium monosodium salt at 450 nm. The cell numbers were reduced by inhibitor treatment in a concentration-dependent manner (Figure 5A). IC50 values are shown in the graph.

To examine the effects of these inhibitors on M-phase progression, cells were treated with inhibitors for 24 h at a concentration close to IC50. Synchronized cells at the G2/M border were released into the M phase, and each cell was examined for mitotic sub-phases, based on microtubule and chromosome morphologies. Without inhibitors, 28% of the cells entered the M phase (Figure 5B, right, DMSO), and 45% of M-phase cells progressed to cytokinesis (Figure 5B, left, dimethyl sulfoxide [DMSO]). Although only 22% of solvent (dimethyl sulfoxide) control cells did not align their chromosomes and were classified as P/PM, more than 45% of cells did not align their chromosomes upon treatment with 5 µM PQ401, 10 µM OSI-906, and 6 µM NVP-ADW742 (Figure 5B, left, P/PM). In addition, the decrease in the mitotic index suggests that mitotic entry was inhibited by these inhibitors (Figure 5B, right). These results suggest that IGF1R inhibitors delay M-phase progression by inhibiting chromosome alignment, and/or mitotic entry.

We next examined the effects of these inhibitors on downstream signaling of IGF1R by detecting phosphorylation of IGF1R at the Tyr1135 and Tyr1136 autophosphorylation sites. While serum-starved cells did not show phosphorylation, addition of IGF1 after starvation caused phosphorylation of IGF1R (Figure 5C, pIGF1R). In addition, phosphorylation of AKT downstream of IGF1R was also clearly observed, indicating activation of the IGF1/IGF1R signaling pathway upon IGF1 treatment. Treatment of cells with OSI-906 and NVP-ADW742 reduced phosphorylation, confirming inhibition of IGF1R signaling at these concentrations. Surprisingly, PQ401 did not inhibit IGF1R phosphorylation at the concentrations used here. M-phase delay in PQ401-treated cells, therefore, may be caused by the off-target effects of PQ401 at this concentration. However, the effects of the other two inhibitors on the M phase can be attributed to inhibition of IGF1R. 

To investigate the direct effect of IGF1R inhibition on M-phase progression, cells were treated with the IGF1R inhibitors during the last 1 h of RO-3306 treatment and the following 1-h release from RO-3306 treatment. Although the effects of this 2-h treatment with these inhibitors were weaker than the effects of the 2-day treatment, inhibitor treatment significantly increased the number of prophase/prometaphase cells (Figure 5D, left, P/PM), indicating that IGF1R inhibition delays M-phase progression. These results suggest that IGF1R directly regulates the M phase. In contrast to 2-day treatment, 2-h treatment did not reduce the mitotic index, suggesting that IGF1R activity is not needed in 2 h around mitotic entry to allow cells to accumulate in M phase.

To determine which sub-phases were affected by IGF1R inhibitors, time-lapse imaging was performed for RO-3306-synchronized cells in the presence of OSI-906 (Figure 6A). In addition to prophase/prometaphase (P/PM, 17 min in DMSO, 21 min in OSI-906), metaphase (M) was also prolonged (31 min in DMSO, 57 min in OSI-906). As a result, anaphase onset was delayed upon IGF1R inhibition (Figure 6B). These results suggest that IGF1R inhibitors delay M-phase progression by extending prophase/prometaphase and metaphase. As described above, IGF1R knockdown only slightly prolonged metaphase (Figure 2B). Partial knockdown may have caused this difference. Anaphase onset is regulated by the spindle assembly checkpoint (SAC) [23]. Until all chromosomes are properly connected with microtubules emanating from two opposite poles, the SAC is activated and anaphase onset is prevented. The delay in anaphase onset after the prolongation of metaphase suggests activation of the SAC in IGF1R-inhibited cells. Prolongation of prophase/prometaphase implies a defect in mitotic spindle formation. Thus, in metaphase, although all chromosomes were mostly aligned at the cell equator, the mitotic spindle would have some defect that reduces inter-kinetochore tension and activates SAC [23]. To explore the mechanism underlying the prolongation of metaphase, we performed cold treatment of IGF1R-inhibited cells to examine microtubule stability of the mitotic spindle. It is known that cold treatment depolymerizes microtubules and that kinetochore microtubules in metaphase spindle are relatively stable against cold treatment [24]. Cold treatment reduced microtubule fluorescence intensities; however, no difference was observed between solvent control (DMSO) and OSI-906-treated cells (Appendix A). This suggests that kinetochore–microtubule attachment may remain unaffected by OSI-906 treatment. Further study will be required to understand how metaphase duration was prolonged via SAC activation in IGF1R-inhibited cells.

### 2.4. Aurora B Inhibitor Potentiates IGF1R Inhibitor-Induced Suppression of Cell Proliferation

IGF1R is frequently overexpressed in some types of cancer [25] and, in some cases, is required for oncogenic transformation [26,27,28], making it an attractive target for cancer chemotherapy. Because combination therapy with the other modes can effectively decrease the dose of anticancer drugs, we explored the substances that enhance IGF1R inhibitor-induced suppression of cell proliferation. In this study, we found that IGF1R inhibitor delays M-phase progression. We, therefore, examined the effects of a combination of IGF1R inhibitor with the Aurora B inhibitor ZM447439 on cell proliferation. Cells were treated with OSI-906, ZM447439, or a combination of the two for 48 h, and the number of cells was then evaluated by WST-8 assay. OSI-906 treatment at 3 µM decreased the number of cells to 52% (Figure 7A). Although 1 µM ZM447439 alone did not decrease cell numbers, it did significantly enhance the suppressive effect of OSI-906 on cell proliferation. Growth curve in the presence of inhibitors clearly demonstrates that ZM447439 potentiates OSI-906-induced suppression of cell proliferation (Figure 7B), suggesting that Aurora B inhibition may potentiate the suppressive effect of IGF1R inhibition on cell proliferation.

Aurora B inhibition is known to induce M-phase defects, including defect in chromosome alignment and SAC inhibition [29]. Thus, we explored whether SAC inhibition was sufficient for this synergistic effect. We first examined whether SAC is activated in IGF1R-inhibited cells by using the Mps1 inhibitor AZ3146 [30] (Figure 7C). Cells were synchronized with RO-3306 and released in the presence of OSI-906, AZ3146, or the combination of both inhibitors for 1 h. AZ3146 treatment alone caused an increase in the number of cytokinesis cells compared with DMSO controls, indicating an acceleration of M-phase progression and confirming inhibition of the SAC in this experimental condition. The number of prophase/prometaphase cells increased in response to OSI-906 treatment, indicating M-phase delay. As expected, combination treatment decreased the number of prophase/prometaphase and metaphase cells and increased the number of cytokinesis cells compared with OSI-906 alone, indicating that SAC inhibition canceled OSI-906–induced M-phase delay and suggesting SAC activation in IGF1R inhibitor-treated cells. Since prophase/prometaphase was prolonged by OSI-906, we speculate that IGF1R inhibition may cause a defect in mitotic spindle, indirectly leading to SAC activation. Next, OSI-906 was combined with AZ3146 in the cell proliferation assay; however, this combination did not produce synergistic effects (Figure 7D), suggesting that SAC inhibition alone is insufficient to potentiate the suppressive effect of OSI-906 on cell proliferation. 

Interestingly, multinucleated cells were frequently observed in cells treated with the combination of OSI-906 and ZM447439, but not AZ3146 (Figure 7E, Appendix A), suggesting cytokinesis failure. Moreover, a severe defect in chromosome alignment was observed in cells treated with this combination for only 2 h (Figure 7F). These results raise the possibility that Aurora B inhibition-induced defects in M phase, including defects in chromosome alignment, SAC, and cytokinesis, may potentiate the effects of OSI-906. Moreover, the combination of OSI-906 and ZM447439 caused cell death (Figure 7E, Appendix A), suggesting that the suppression of cell proliferation by the combination may be caused by the delay in M-phase progression and induction of cell death.

## 3. Discussion

M-phase progression is regulated by many kinases, including CDK1, Aurora kinases, and PLK1. However, fewer reports of the involvement of receptor-type tyrosine kinases in M-phase regulation have been published. In the present study, we found that the IGF1/IGF1R signal plays a role in M-phase progression. IGF1R inhibitors delayed M-phase progression possibly via a defect in spindle formation and indirect SAC activation. This delay in M-phase progression may contribute to IGF1R inhibitor-induced suppression of cancer cell proliferation. Furthermore, the Aurora B inhibitor enhanced the suppressive effect of IGF1R inhibitor on cell proliferation, raising the potential for a new therapeutic approach.

In this study, IGF1R knockdown delayed M-phase progression. Because it takes two to three days to achieve knockdown, IGF1R knockdown would affect M-phase regulators at the transcriptional and translational levels and thereby affect M-phase progression. However, we observed no change in several M-phase regulators at the transcriptional and translational levels. We also investigated the effect of IGF1R inhibitors at short treatment times. IGF1R inhibition by these inhibitors produced results similar to those of siRNA in terms of M-phase delay, confirming that IGF1R inhibition is responsible for the M-phase delay observed here. Interestingly, only 2 h of treatment delayed M-phase progression, suggesting that IGF1R inhibition directly affects M-phase progression. We therefore concluded that IGF1R may participate in M-phase progression during cell division.

Time-lapse imaging revealed that IGF1R inhibition delayed chromosome alignment and anaphase onset. This delay was canceled by treatment with the Mps1 inhibitor AZ3146. Mps1 phosphorylates Knl1, a component of the KMN network, which supports assembly of the mitotic checkpoint complex by providing docking sites for SAC proteins [31]. Mps1 is therefore an essential component of the SAC, and AZ3146 prevents activation of the SAC by inhibiting Mps1 kinase [30]. Even when not all chromosomes were properly aligned at the cell equator, cells prematurely exited mitosis upon AZ3146 treatment. Thus, the cancellation of IGF1R inhibitor-induced M-phase delay suggests that SAC is involved in the delay in the onset of anaphase. Considering that prophase/prometaphase was prolonged by IGF1R inhibition, we speculate that IGF1R inhibition causes defects in spindle formation or function, resulting in indirect SAC activation and delay in the onset of anaphase. Cold treatment of IGF1R-inhibited cells resulted in no difference in the fluorescence intensity of microtubules between control and OSI-906-treated cells, suggesting that kinetochore–microtubule attachment may be unaffected by OSI-906 treatment. Further study will be required to understand how IGF1R participates in spindle formation or function.

FoxM1 is a transcription factor crucial for cell-cycle progression at the G1/S and G2/M transitions [19]. Its target genes include CDK1, cyclin B1, Aurora A, Aurora B, PLK1, CDC25B, CENP-B, and CENP-F. Knockdown of FoxM1 results in a delay of mitotic entry and M-phase defects, such as chromosome misalignment and cytokinesis failure, leading to the formation of aneuploid and polyploid cells [19]. Expression levels are dependent on the cell cycle; expression increases in the G2/M phase in the human foreskin fibroblastic cell line hTERT-BJ1 [22] and degrades during mitotic exit [32]. FoxM1 expression is negatively regulated by the FOXO transcription factor FOXO3a [33]. AKT phosphorylates the FOXO transcription factor FOXO3a. By this phosphorylation, FOXO3a is negatively regulated; FOXO3a is excluded from the nucleus and degraded via ubiquitination [34]. Thus, the activation of AKT increases FoxM1 expression. IGF1/IGF1R signal phosphorylates IRS1, activating PI3K and thereby also activating AKT. We observed a striking increase in the phosphorylation of AKT after IGF1 treatment; however, no increase in FoxM1 levels was observed. For transcriptional activation, Raf/MEK/MAPK-stimulated nuclear translocation is required [22]. IR facilitates the nuclear localization of FoxM1 by triggering the MEK/ERK pathway, leading to cell proliferation via the expression of M-phase regulators. Because IGF1R is highly homologous to IR, we analyzed the effect of IGF1R knockdown on FoxM1. However, the nuclear localization of FoxM1 was only partially inhibited by IGF1R inhibition. In addition, protein levels of M-phase regulators were not changed. Therefore, the contribution to FoxM1 transcriptional activity may differ between IR and IGF1R. Taking these findings together, we conclude that IGF1R inhibition-induced M-phase delay is not caused by the inhibition of FoxM1 activity.

IGF1R is frequently overexpressed in a wide variety of human cancers, and IGF1R signaling may play a role in oncogene-induced cancer transformation in certain cancers [35]. IGF1/IGF1R signaling is therefore an attractive therapeutic target for cancer. One major problem in cancer chemotherapy is the development of resistance to anticancer drugs. Interestingly, the IGF1R inhibitor picropodophyllin reportedly reverses resistance to cisplatin and taxol at the early stages [36], and combination therapy may improve drug sensitivities. In addition, IGF1R is expressed in normal tissues, and many IGF1R inhibitors cross-react with IR. For this reason, combination therapy has advantages in reducing the required dose of IGF1R inhibitors. Combination-therapy approaches targeting IGF1R and its downstream signaling pathways have been widely applied [37]. In the present study, inhibition of Aurora B enhanced the suppressive effect of IGF1R inhibitor on cell proliferation. The concentration of the Aurora B inhibitor used in this study was too low to affect cell proliferation. This means that these combined effects are not just additive.

How does Aurora B inhibition enhance the suppressive effects of IGF1R inhibitors? We found that IGF1R inhibitor-induced M-phase delay was accompanied by SAC activation. Aurora B is required for the recruitment of Mps1 [38,39,40]; therefore, Aurora B inhibition inactivates SAC, causing precocious onset of anaphase. Thus, we had hypothesized that SAC inhibition was sufficient to produce a synergistic effect. However, the combination of OSI-906 with the Mps1 inhibitor AZ3146 did not show any synergistic effects, indicating that SAC inhibition alone is insufficient to produce synergistic effects. In addition to the SAC inhibition, Aurora B inhibition causes a defect in chromosome alignment [29]. Thus, chromosome alignment may be the common target of IGF1R inhibition and Aurora B inhibition. In addition, Aurora B plays a role in abscission checkpoint [41] and Aurora B inhibition results in cytokinesis failure via furrow regression in cells having a chromosome bridge. We observed a severe defect in chromosome alignment upon the use of this combination. Given that Aurora B inhibition prevents SAC activation, the combination may cause severe chromosome segregation errors via precocious onset of anaphase and produce multinucleated cells via cytokinesis failure. Further study to clarify the mechanisms behind the effects of this combination should provide a new strategy for cancer chemotherapy using IGF1R inhibitors and also provide insight into the role of IGF1R in M-phase progression.

## 4. Materials and Methods

### 4.1. Cells

The human cervix adenocarcinoma cell line HeLa S3 (Japanese Collection of Research Bioresources, Osaka, Japan) was grown in Dulbecco’s modified Eagle’s medium supplemented with 20 mM 4-(2-hydroxyethyl)-1-piperazine ethanesulphonic acid (HEPES)-NaOH (pH 7.4) and 5% FBS.

### 4.2. Chemicals

The IGF-1R inhibitors PQ401 (2768, Tocris Bioscience, Ellisville, MO, USA), OSI-906 (07333, LKT Laboratories, St. Paul, MN, USA), NVP-ADW742 (S1088, Selleck Chemicals, Houston, TX, USA), the Aurora B inhibitor ZM447439 (JS Research Chemicals Trading, Wedel, Germany), the PLK1 inhibitor BI12536 (A10134, AdooQ Bioscience, Irvine, CA, USA), and the MPS1 inhibitor AZ3146 (A11170, AdooQ Bioscience) were used. For cell-cycle synchronization to M phase, the reversible CDK1 inhibitor RO-3306 (S7747, Selleck Chemicals) was used. These inhibitors were dissolved in DMSO (Nacalai Tesque, Kyoto, Japan). Recombinant human IGF1 was purchased from Peprotech (100-11, London, UK).

### 4.3. siRNA

To knockdown IGF1R, siRNA against IGF1R was used. siIGF1R #1 (5’-CAAUGAGUACAACUACCGCUU-3’) was synthesized by Merck (Darmstadt, Germany). siIGF1R #2 (SASI_Hs01_00126195) and non-targeting control siRNA (SIC-001, Merck) were purchased from Merck. siRNA was transfected using the Lipofectamine 2000 reagent (Thermo Fisher Scientific, Waltham, MA, USA) into HeLa S3 cells at 25 nM. At 48 h after the transfection, knockdown of the target proteins was verified using Western blotting.

### 4.4. Antibodies

The following antibodies were used for immunofluorescence (IF) and Western blotting (WB): mouse monoclonal anti-FoxM1 (WB, 1:1000–2000; IF, 1:200; sc-376471, G-5, Santa Cruz Biotechnology, Dallas, TX, USA), anti-phospho p44/42 MAPK (Erk1/2) (Thr202/Tyr204) (WB, 1:1000; #9106, Cell Signaling Technology, Danvers, MA, USA), anti-AKT (Pan) (WB, 1:1000; 40D4, #2920, Cell Signaling Technology), and anti-Plk (WB, 1:1000; F-8, sc-17783, Santa Cruz Biotechnology) antibodies, rat monoclonal anti-α-tubulin (WB, 1:2000; IF, 1:800; YOL1/34, MCA78G, Bio-rad, Hercules, CA, USA) antibody, rabbit monoclonal anti-IGF-1 receptor β (WB, 1:1000; D23H3, #9750, Cell Signaling Technology), anti-phospho-IGF-1 receptor β (Tyr1135/1136)/IR β (Tyr1150/1151) (WB, 1:1000; #3024, Cell Signaling Technology), anti-phospho-AKT (Ser473) (WB, 1:1000; D9E, #4060, Cell Signaling Technology), anti-phospho-Src family (Tyr416) (WB, 1:500; #2101, Cell Signaling Technology) antibodies, rabbit polyclonal anti-ERK2 (WB, 1:500–1000; C-14, sc-154, Santa Cruz Biotechnology), and anti-cyclin B1 (WB, 1:2000; H-433, sc-752, Santa Cruz Biotechnology) antibodies.

As the secondary antibodies for IF staining, Alexa Fluor 488-labeled donkey anti-mouse immunoglobin G (IgG) (1:800; A21202), Alexa Fluor 488–labeled donkey anti-rat IgG (1:800; A21208), and Alexa Fluor 555–labeled goat anti-rat IgG (1:800; A21434) antibodies were purchased from Thermo Fisher Scientific (Waltham, MA, USA) and used at a dilution of 1:800. For WB analysis, horseradish peroxidase (HRP)-conjugated donkey anti-mouse IgG (715-035-151), donkey anti-rabbit IgG (711-035-152), and donkey anti-rat IgG (712-035-153) antibodies were purchased from Jackson ImmunoResearch (West Grove, PA, USA) and used at a dilution of 1:4000.

### 4.5. Immunofluorescence Microscopy

Immunofluorescence staining was performed, as described previously [42,43]. In brief, cells were fixed with 4% formaldehyde in phosphate-buffered saline (PBS) at room temperature for 20 min. The fixed cells were permeabilized with PBS supplemented with 0.1% saponin and 3% bovine serum albumin and then incubated with primary and secondary antibodies for 1 h each in the same solution. DNA was stained with 1 µM Hoechst 33342 during the incubation with a secondary antibody. A fluorescence microscope (IX-83, Olympus, Tokyo, Japan) equipped with a 20× 0.45 NA and a 40× 0.75 NA objective lens (Olympus) was used to capture fluorescence images. Hoechst 33342, Alexa Fluor 488, and Alexa Fluor 555 fluorescence were detected through a U-FUNA (360–370 nm excitation, 420–460 nm emission), U-FBNA (470–495 nm excitation, 510–550 nm emission), and U-FRFP (535–555 nm excitation, 570–625 nm emission) filter cubes, respectively. For Figure 7E and Appendix A, cell images were captured using the Floid^TM^ Cell Imaging Station (Thermo Fisher Scientific). Fluorescence images were edited using ImageJ (National Institutes of Health, Bethesda, MD, USA), Photoshop CC, and Illustrator CC software (Adobe, San Jose, CA, USA).

### 4.6. Western Blotting

Cells were solubilized in an SDS-sample buffer supplemented with phosphatase and protease inhibitors [50 mM NaF, 20 mM β-glycerophosphate, 10 mM Na_3_VO_4_, 2 µg/mL aprotinin, 0.8 µg/mL pepstatin, 2 µg/mL leupeptin, 0.5 mM EGTA, and 2 mM phenylmethylsulfonyl fluoride (PMSF)] and incubated at 40 °C for 20 min. Proteins that were separated by SDS-polyacrylamide gel electrophoresis were transferred onto polyvinylidene difluoride membranes (Pall Corporation, Port Washington, NY, USA). After being blocked with Blocking One (Nacalai Tesque, Kyoto, Japan), the membrane was incubated for 1 h at room temperature or overnight at 4 °C with primary and secondary antibodies diluted in tris-buffered saline supplemented with 5% Blocking One and 0.1% Tween20. For sequential re-probing of the membranes with various antibodies, HRP was inactivated by 0.1% NaN_3_. Chemi-Lumi One L (07880-70, Nacalai Tesque) and Clarity (#1705061, Bio-Rad) were used as the chemiluminescence substrate, and chemiluminescence was detected using the image analyzer ChemiDoc XRSplus (Bio-Rad).

### 4.7. Cell-Cycle Synchronization

To analyze the effects of inhibitors and knockdown of IGF1R on M-phase progression, cells were arrested at the G2/M border by treating cells with the CDK1 inhibitor RO-3306 at 6 µM for 20 h. To release cells from the G2/M arrest, the cells were washed with prewarmed PBS supplemented with Ca^2+^ and Mg^2+^ [PBS(+)] more than three times in a water bath at 37 °C, and incubated in prewarmed medium for 50–60 min. The cells were then fixed with 4% formaldehyde in PBS for 20 min at room temperature. The fixed cells were stained for α-tubulin and DNA and classified into four categories, prophase/prometaphase (P/PM), metaphase (M), anaphase/telophase (A/T), and cytokinesis (Cyto), based on the microtubule and chromosome morphologies. The percentage of each category was calculated.

### 4.8. Time-Lapse Imaging

HeLa S3 cells were seeded in a 24-well plate and transfected with either siCtrl or siIGF1R #2. After 28 h, cells were treated with 6 µM RO-3306 for 20 h. After being washed with prewarmed PBS(+) on a water bath, 0.1 µM Hoechst 33342 was added to the culture, and time-lapse imaging was performed using the Operetta imaging system (PerkinElmer, Waltham, MA, USA). Time-lapse images of bright-field and Hoechst 33342 were acquired every 5 min for 140 min in a live-cell chamber at 37 °C in 5% CO_2_ [44,45,46]. The duration of each category, such as P/PM, M, A/T, and blebbing cells, was determined.

### 4.9. Proliferation Assay

Cell viability was determined with water-soluble tetrazolium salt, 2-(2-methoxy-4-nitrophenyl)-3-(4-nitrophenyl)-5-(2,4-isulfophen-yl)-2H-tetrazolium, monosodium salt (WST-8, a Cell Counting Kit-8, 343-07623, Dojindo, Kumamoto, Japan). Cells were seeded at 2–4 × 10^3^ per well in 96-well plate. The next day, cells were cultured with the IGF1R inhibitors, PQ401 (0.3–10 µM), OSI-906 (1–30 µM), and NVP-ADW742 (1–30 µM) for 48 h. DMSO was added to the culture at 0.1% as a solvent control. To examine combination effects, 3 µM OSI-906, 1 µM ZM447439, and 2 µM AZ3146 were used. After incubation with WST-8 for 3–4 h, the absorbance of formazan dye was measured at 450 nm. The ratios of the absorbance of inhibitor-treated cells to that of control cells were calculated.

To assess cell proliferation, cells were seeded at 2 × 10^4^ cells/well in a 24-well plate; the next day, 3 µM OSI-906, 1 µM ZM447439, or DMSO was added into the culture medium. Every 24 h, cells were washed with PBS(−) and trypsinized. After staining with 0.4% (*w*/*v*) trypan blue in PBS(−), the number of trypan blue-negative cells was counted as living cells using a hemocytometer.

### 4.10. Real-Time PCR

HeLa S3 cells were transfected with siCtrl, siIGF1R #1, or #2, and 40 h later, cells were treated with 4 mM thymidine (T1895-10G, Merck) for 24 h. Subsequently, cells were washed and then cultured with prewarmed fresh medium for 8 h. Cells were lysed, treated with DNaseI, and reverse-transcribed into cDNA using the TaqMan Fast Advanced Cells-to-Ct kit (Thermo Fisher Scientific, Waltham, MA, USA) according to the manufacturer’s instructions. Quantitative PCR was performed using TaqMan probe-based gene expression analysis and QuantStudio 1 (Thermo Fisher Scientific). The primers and probes purchased from Thermo Fisher Scientific were as follows: cyclin B1 (CCNB1, Hs01030099_m1), Aurora A (AURKA, Hs01582072_m1), Aurora B (AURKB, Hs00945858_g1), CDC25B (CDC25B, Hs01582335_m1), GAPDH (GAPDH, Hs02786624_g1), CENPB (CENPB, Hs00374196_s1), CENPF (CENPF, Hs01118845_m1), CDK1 (CDK1, Hs00938777_m1), PLK1 (PLK1, Hs00983227_m1), and β-actin (ACTB, Hs01060665_g1). Expression of the housekeeping gene ACTN or GAPDH was used as internal control. Relative expression was analyzed by the ∆∆Ct method using Ct values.

### 4.11. Statistics

Statistical analysis was performed with the Statcel 4 add-in program for Microsoft Excel (OMS Publishing, Tokorozawa, Japan) using the results of more than three independent experiments. For analysis between two groups, an F test was used to determine the homogeneity of variance. Welch’s t-test was used for groups with unequal variance. For analysis among more than three independent groups, the Bartlett test was used to determine the homogeneity of variance. For groups with equal variance, data were analyzed by one-way analysis of variance (ANOVA) and then by the Tukey–Kramer multiple comparisons test. For groups with unequal variance, Kruskal–Wallis was used, followed by the Steel–Dwass test. A *p* value of less than 5% was considered statistically significant.

## Figures and Tables

**Figure 1 ijms-21-01058-f001:**
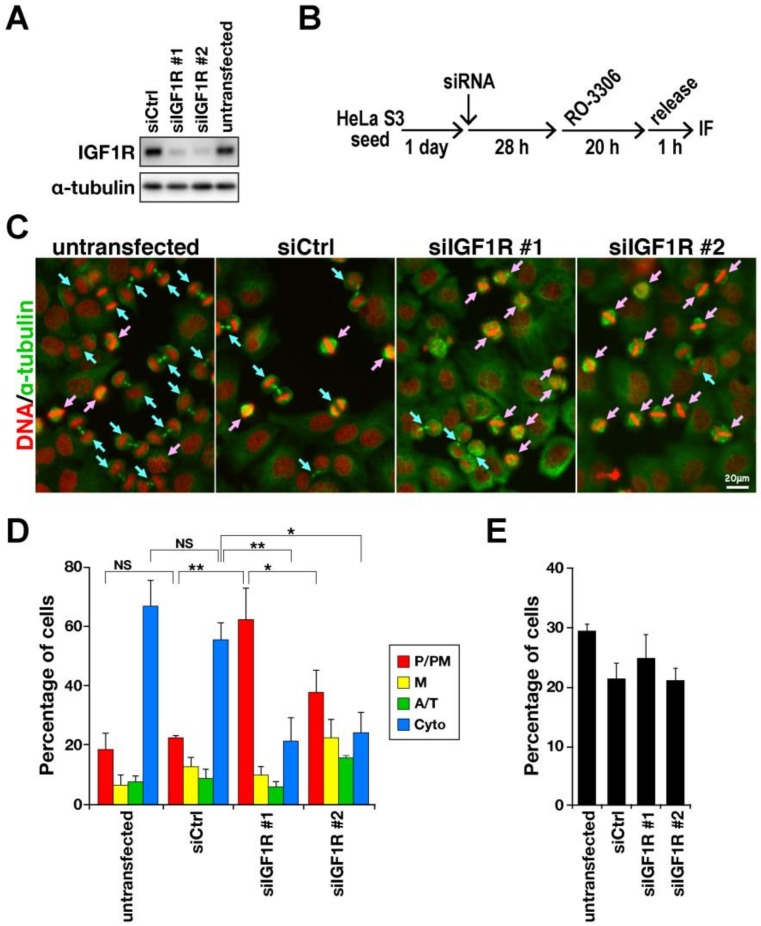
IGF1R knockdown delays M-phase progression. (**A**) HeLa S3 cells were transfected with non-targeting siRNA (siCtrl) and siRNAs targeting IGF1R (#1, #2), and 48 h later, whole-cell lysates were analyzed by western blotting analysis with indicated antibodies. (**B**–**E**) HeLa S3 cells were transfected with non-targeting siRNA (siCtrl) and siRNAs targeting IGF1R (#1, #2). After 28 h, cells were treated with 6 µM RO-3306 for 20 h. After washing cells, the cells were incubated with prewarmed medium for 1 h and fixed, followed by immunofluorescence staining for DNA (red) and α-tubulin (green). M-phase cells were classified into four groups: cells in prophase and prometaphase (P/PM), metaphase (M), anaphase and telophase (A/T), and cytokinesis (Cyto). The ratio of M-phase cells among all cells was determined by examining more than 979 cells in each experiment. (**B**) A schematic depiction of the synchronization method is shown. (**C**) Representative images are shown. Pink and blue arrows indicate cells before and after anaphase onset, respectively. Scale bar, 20 µm. (**D**) The percentages of cells in each group are plotted as mean ± SD calculated from three independent experiments (*n* > 206 in each experiment). The asterisk indicates significant differences using Tukey–Kramer test. * *p* < 0.05, ** *p* < 0.01, NS, not significant. (**E**) The mitotic index is plotted as mean ± SD. There was no significant difference (Tukey–Kramer test).

**Figure 2 ijms-21-01058-f002:**
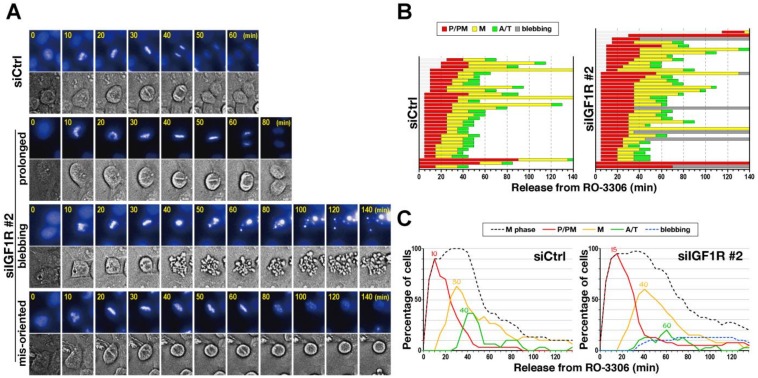
Delay in chromosome alignment and anaphase onset. HeLa S3 cells were transfected with control siRNA (siCtrl) or siIGF1R (siIGF1R #2), and 28 h later, cells were treated with 6 µM RO-3306 for 20 h. Cells were released in the presence of 0.1 µM Hoechst 33342 to visualize DNA. M-phase progression was monitored every 5 min for 140 min by time-lapse imaging. (**A**) Representative images of cells showing normal M-phase, delayed progression, blebbing, and misorientation of the mitotic spindle are shown. (**B**) The duration of each mitotic sub-phase—prophase and prometaphase (P/PM, red), metaphase (M, yellow), anaphase and telophase (A/T, green), and blebbing cells (bleb, gray) for individual cells are shown (siCtrl, *n* = 32; siIGF1R, *n* = 40). (**C**) The percentages of M-phase cells (black), prophase and prometaphase cells (red), metaphase (orange), anaphase and telophase cells (green), and blebbing cells (blue) at the indicated times are plotted. The respective peak times for the ratios of sub-phases are shown in the graph.

**Figure 3 ijms-21-01058-f003:**
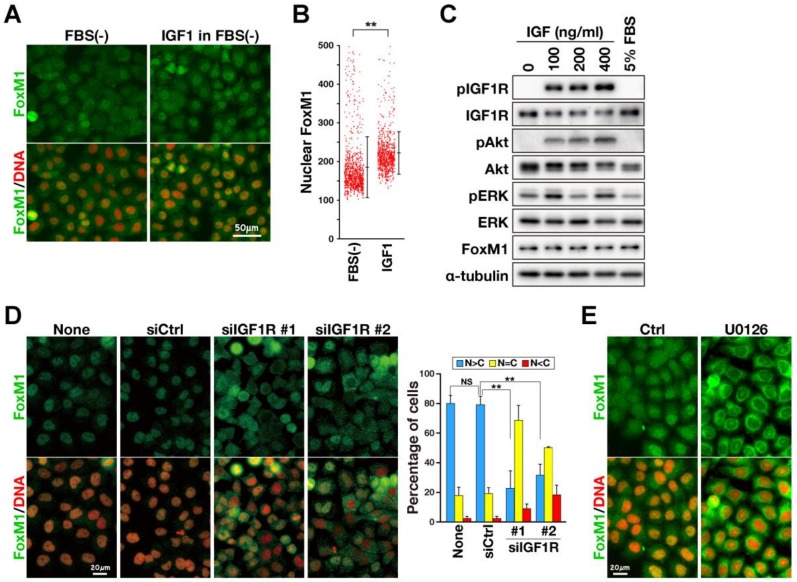
IGF1R signal partially induce nuclear localization of FoxM1. (**A**, **B**) HeLa S3 cells were cultured with FBS-free medium for 24 h and then incubated in the presence of 0.1 µg/mL IGF1 for 24 h. The cells were fixed and stained for FoxM1 (green) and DNA (red). (**A**) Representative images are shown. Scale bar, 50 µm. (**B**) The fluorescence signals were analyzed by an image analyzer. Fluorescence intensities of nuclear FoxM1 in individual cells were plotted with the mean ± SD calculated from one experimental result *(n* > 930 in each experiment). Two independent experiments were performed and similar results were obtained. The asterisk indicates significant differences using Welch’s *t*-test. ** *p* < 0.01. (**C**) HeLa S3 cells were starved for 24 h and then treated with 100, 200, and 400 ng/mL of IGF1 for 3 h. Whole-cell lysates were prepared and analyzed by western blots with the indicated antibodies. (**D**) HeLa S3 cells were transfected with non-targeting siRNA (siCtrl) and siRNAs targeting IGF1R (#1, #2). After 48 h, cells were fixed with formaldehyde and staining for FoxM1 (green) and DNA (red). Cells were classified into three categories: (*N* > *C*), fluorescence in the nucleus is higher than that in the cytoplasm; (*N* = *C*), similar fluorescence intensity throughout the nucleus and cytoplasm; (*N* < *C*), fluorescence in the nucleus was lower than that in the cytoplasm. Results represent mean ± SD from three independent experiments (*n* > 233 in each experiment). The asterisk indicates significant differences using a Tukey–Kramer test. ** *p* < 0.01, NS, not significant. (**E**) HeLa S3 cells were treated with or without U0126 in serum-free culture medium containing 100 ng/mL of IGF1 for 24 h, fixed and stained for FoxM1 (green) and DNA (red). Scale bar, 20 µm.

**Figure 4 ijms-21-01058-f004:**
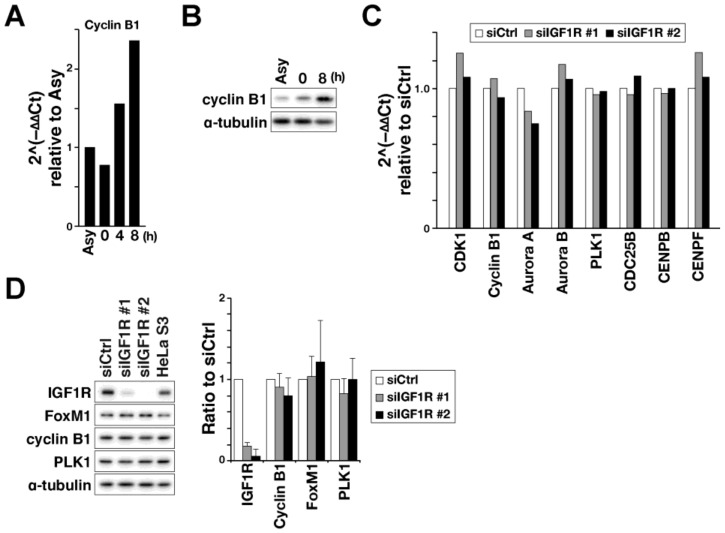
FoxM1 target genes are not suppressed in IGF1R knockdown cells. (**A**, **B**) HeLa S3 cells were treated with 4 mM thymidine for 24 h, washed with prewarmed PBS(–), and then cultured with prewarmed medium for 4 or 8 h. (**A**) Asynchronized cells (Asy), cells treated with thymidine (0 h) and those released from thymidine treatment for 4 or 8 h were collected and analyzed for cyclin B1 mRNA by real-time PCR. (**B**) Asynchronized cells (Asy) and cells treated with thymidine (0 h) and those released from thymidine treatment for 8 h were collected and subjected to western blot analysis using indicated antibodies. (**C**) HeLa S3 cells were transfected with siCtrl, or siIGF1R #1 or #2 and treated 40 h later with 4 mM thymidine for 24 h. Cells were then washed, cultured with prewarmed medium for 8 h, and collected for real-time PCR analysis using TaqMan probes for M-phase regulators indicated in the figure. Fold changes in mRNA levels are expressed relative to siCtrl. (**D**) HeLa S3 cells were transfected with siCtrl or siIGF1R #1 or #2 and treated 16 h later with 4 mM thymidine for 24 h. Cells were then washed, cultured with prewarmed medium for 8 h, and collected for western blot analysis. Protein levels were quantified by measuring band intensities and relative ratios to siCtrl are plotted as mean ± SD calculated from three independent experiments. There was no significant difference (Steel–Dwass test).

**Figure 5 ijms-21-01058-f005:**
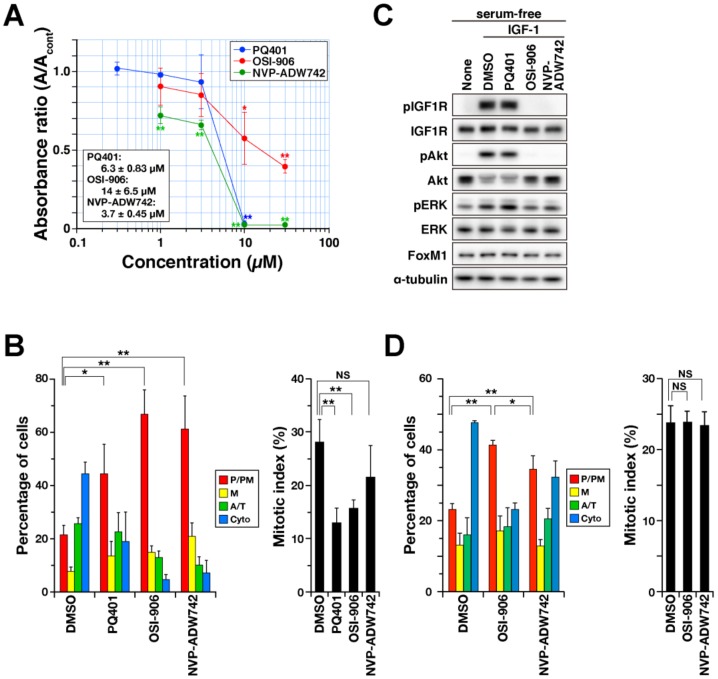
IGF1R inhibitors delays M-phase progression. (**A**) HeLa S3 cells were treated with PQ-401, OSI-906, and NVP-ADW742 at the indicated concentrations for 48 h, and the cell viability was assayed using a water-soluble tetrazolium salt (WST-8). Relative values of inhibitor-treated cells were calculated as a ratio of absorbance to solvent control (dimethyl sulfoxide, DMSO). The means ± SD of three independent experiments are shown. IC50 was calculated in inhibitor-treated cells in each experiment, and the mean ± SD is shown in the graph. Asterisks indicate statistical significance (* *p* < 0.05, ** *p* < 0.01) calculated using Scheffe’s F test. (**B**) HeLa S3 cells were treated with the IGF1R inhibitors (5 µM PQ401, 10 µM OSI-906; 6 µM NVP-ADW742) or solvent control (DMSO) for 24 h. During the last 20 h of inhibitor treatment, cells were treated with 6 µM RO-3306 in the presence of inhibitors. Cells were then washed and cultured with prewarmed medium for release from the G2/M arrest. After 50 min, cells were fixed with 4% formaldehyde and stained for α-tubulin and DNA. On the basis of α-tubulin and DNA morphologies under a microscope, M-phase cells were classified into four groups: prophase/prometaphse (P/PM), metaphase (M), anaphase/telophase (A/T), and cytokinesis (Cyto). The percentages of cells are plotted as the mean ± SD, calculated from three independent experiments (n > 161 in each experiment). The Tukey–Kramer multiple comparisons test was used to calculate *p* values (* *p* < 0.05, ** *p* < 0.01). (C) HeLa S3 cells were serum-starved for 24 h and then treated with the inhibitors (5 µM PQ401, 10 µM OSI-906; 6 µM NVP-ADW742) for 1 h. During the last 30 min of inhibitor treatment, cells were stimulated with 0.1 µg/mL of IGF-1. Whole-cell lysates were analyzed using western blot analysis with the indicated antibodies. (**D**) HeLa S3 cells were synchronized with RO-3306 and M-phase progression was assayed as described in (**B**), except for the treatment time; during the last 1 h of RO-3306 treatment and 1-h release from RO-3306 treatment, cells were continuously treated with the IGF1R inhibitors (10 µM OSI-906; 6 µM NVP-ADW742). The percentages of cells are plotted as the mean ± SD, calculated from three independent experiments (*n* > 213 in each experiment). The Tukey–Kramer multiple comparisons test was used to calculate *p* values (* *p* < 0.05, ** *p* < 0.01, NS, not significant).

**Figure 6 ijms-21-01058-f006:**
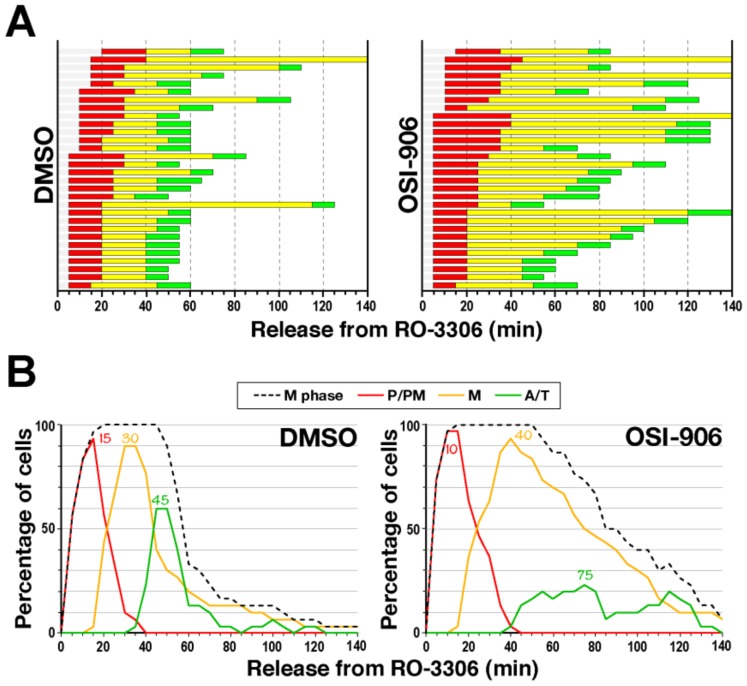
IGF1R inhibitor extends prophase/prometaphase and metaphase. HeLa S3 cells were synchronized with RO-3306 by incubating for 20 h. Just after the release from RO-3306 treatment, time-lapse imaging was performed as described in Figure 2. During the last 1 h of incubation with RO-3306 and the following release, cells were treated with 10 µM OSI-906. (**A**) The durations of each mitotic sub-phase [P/PM (red), M (yellow), A/T (green)] for individual cells are shown (siCtrl, *n* = 30; siIGF1R, *n* = 30). (**B**) The percentages of M-phase cells (black), P/PM (red), M (orange), and A/T cells (green) at the indicated times are plotted. The respective peak times for the ratios of sub-phases are shown in the graph.

**Figure 7 ijms-21-01058-f007:**
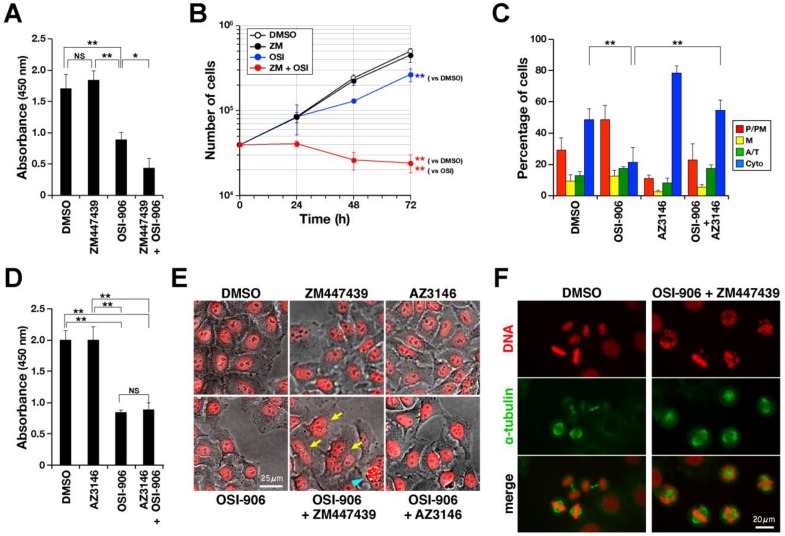
IGF1R inhibitor synergizes with the Aurora B inhibitor to suppress cell proliferation. (**A**) HeLa S3 cells were treated with 3 µM OSI-906 alone or in combination with 1 µM ZM447439. The cell viability was assessed using WST-8. The mean ± SD of absorbance was calculated from three independent experiments. (**B**) HeLa S3 cells (20,000 cells) were seeded in a 24-well plate. The number of cells was counted every 24 h for three days and plotted. Statistical analysis was performed for comparison of the cell numbers at 72 h. (**C**) HeLa S3 cells were synchronized with RO-3306, and the M-phase progression was examined as described in Figure 5D using a combination of 10 µM OSI-906 and 2 µM AZ3146. The mean ± SD was calculated from three independent experiments (n > 204 in each experiment). (**D**) HeLa S3 cells were treated with 3 µM OSI-906 alone or in combination with 2 µM AZ3146. The cell viability was assessed as described in (**A**). (**E**) At 51 h after the addition of inhibitors or DMSO, cells were stained for DNA with 10 µM Hoechst 33342 (pseudo-colored red). After washing with PBS(–), cell images were captured. Yellow arrows indicate multinucleated cells. Blue arrowhead indicates dead cells. (**F**) HeLa S3 cells were synchronized with RO-3306. During the last 1 h of incubation with RO-3306 and the following 1 h of release, cells were treated with the combination of OSI-906 and ZM447439, and stained for α-tubulin and DNA. Tukey–Kramer multiple comparisons test was used to calculate *p* values (* *p* < 0.05, ** *p* < 0.01, NS, not significant).

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
