# Peer review of "Targeting Insulin-Like Growth Factor 1 Receptor Delays M-Phase Progression and Synergizes with Aurora B Inhibition to Suppress Cell Proliferation"

_ijms, 2020, doi:10.3390/ijms21031058_

Round 1

Reviewer 1 Report

The authors have addressed my suggestions for the manuscript. 

I would have liked to see a graph that actually showed the IGF1R siRNA mean metaphase duration with error bars, but it is possible to see from their live imaging what has happened - it is mostly pro-metaphase delay. 

It is odd that Fig 7C does not show any increase in metaphase incidence upon OSI-906 treatment, since 6A suggests that metaphase is extended (though again the statistics are not shown).

I am happy with the changes to the title and discussion.

Reviewer 2 Report

The authors addressed all my previous concerns. The manuscript improved significantly and in my opinion is now suitable for publication. I commend the authors for their efforts in revising the manuscript.   

This manuscript is a resubmission of an earlier submission. The following is a list of the peer review reports and author responses from that submission.

Round 1

Reviewer 1 Report

This paper tests whether IGFR1 has similar roles to IR in affecting mitosis via FoxM1. Overall, I found the experiments to be appropriately carried out and presented. 

In Figure 1D I would prefer to see the statistical comparison between the Prophase/Prometaphase percentages rather than the cytokinesis percentatges, as the proposed defect is in early mitosis. The Tukey-Kramer test is not ideal for comparing multiple tests to a control - this would be better done with Dunnett's test.

Figure 2B and C do not quantitate the mean time spent in each phase of Mitosis - it looks as if P/PM is longer and maybe Metaphase is longer, but it is very hard to tell just from 2B. 2C is not helping in my opinion. The delays are cumulative, so it is not clear what it means that A/T ends up peaking 20 minutes later - is that primarily from metaphase delay or prophase delay? I would like to see a graph showing the mean timing of each phase.  My guess is that the variation will be too large to allow a confident claim of a metaphase delay, but this is the data that we want to know (particularly given the problems with Figure 6, see below).

I liked Figure 4 - it is useful to see this kind of negative result.

I suggest clarifying line 283 - currently it suggests that IGFR1 activity is not required for anything at M phase entry, whereas the data just suggests that it is not needed in the last two hours to allow cells to accumulate in M phase.

The major problem I have with this paper is the interpretation of Figure 6. The SAC arrests cells in metaphase, not prophase or prometaphase. If loss of IGFR1 activity activated the SAC, you would see an increased percentage of cells in metaphase. This was not observed. 

Instead, a delay in P/PM phase was observed, which would be consistent with a problem in spindle formation, centrosomes, kinetochore capture or several other possibilities, but not SAC activation. Once cells reached metaphase, there was no evidence of a significant problem in moving to anaphase.

Following this, the experiment with Aurora B inhibitor is hard to interpret - the Mps1 inhibitor should block SAC activation like Aurora B inhibition, but Mps1 inhibition seemed to rescue IGFR1 inhibition in Fig 6A, not enhance it as the Aurora inhibitor did in 6B. 

In my opinion, the paper is stronger without Figure 6.  If it is retained, the interpretation should be modified to make it clear that the effect of IGFR1 inhibition has not been identified (and is not a prolonged activation of the SAC), though it is modulated by mitotic regulators such as Aurora B and Mps1.

The discussion should be reworked with the emphasis on the SAC removed. Line 407 appears to be an unjustified assertion - no evidence has been presented of severe chromosome segregation errors or cell death.

The paragraph from lines 350 to 370 does not relate to the data presented in this paper and would be better suited to a review paper.

There were some typographic errors that should be fixed, notably the "Tukey-Kramer test", though turkeys may seem appropriately seasonal. 

Reviewer 2 Report

This manuscript presents an interesting scientific question, which the authors addressed with the appropriate experimental approaches. I have however some reservations in recommending this study for publication as I am not entirely convinced that the main conclusions are actually supported by the results. In my opinion, further experimental data is needed to substantiate the conclusion that inhibition of IGF1R delays cells in mitosis due to SAC activation. Moreover, the manuscript would benefit if the English could be revised. Some lines of thought or ideas are sometimes hard to follow or understand. That being said, this study has potential for publication as long as the following points are addressed:

1- Revise the abstract. The abstract is long and written in a very convoluted way. I advise the authors to focus on the main findings and to be less descriptive. I would suggest something like this:

The insulin-like growth factor 1 receptor (IGF1R) is a receptor-type tyrosine kinase that transduces signals related to cell proliferation, differentiation, and survival. IGF1R expression is often misregulated in tumor cells but the relevance of this for cancer progression remains unclear. Here, we examined the impact of IGF1R inhibition on cell division. We found that siRNA-mediated knockdown of IGF1R from HeLa S3 cells leads to an M-phase delays. Although IGF1R depletion causes partial exclusion of FoxM1, quantitative real-time PCR revealed that the transcription of M-phase regulators is not affected by decreased levels of IGF1R. Moreover, a similar delay in mitosis was observed following a 2h-incubation period with the IGF1R inhibitors OSI-906 and NVP-ADW742. These results suggest that the M-phase delay observed in IGF1R-compromised cells is not caused by altered expression of mitotic regulators. Live-cell imaging revealed that both a prolonged prometaphase and metaphase duration underlie the mitotic delay and this can be abrogated by inhibition of Mps1 with AZ3146, hence suggesting a permanent activation of the Spindle Assembly Checkpoint when IGF1R is inhibited. Furthermore, incubation with the Aurora B inhibitor ZM447439 potentiated the IGF1R inhibitor-induced suppression of cell proliferation, opening new possibilities for more effective cancer chemotherapy.

2- Please revise the introduction. I’ve detected some inaccuracies. For example:

2.1- Line 42 replace “signal” by “signaling”

2.2- Line 44 “In cell division, replicated chromosomes are segregated into two daughter cells by mechanical forces provided by microtubules and actin…” This is not accurate. Depolymerizing microtubules are the primary drivers of chromosome movement during anaphase (please check PMID:31527150).  

2.3- Line 53 “Deregulation of cell division gives rise to asymmetrical chromosome separation, leading to chromosomal instability (CIN), a hallmark of cancer, but the correlation of IGF1R to CIN is unknown.” I suggest removing “but the correlation of IGF1R to CIN is unknown”.

2.4- Line 62 “was prolonged in an IGF1-null uterus” replace by “was prolonged in IGF1-null mice uterine cells.”

2.5- Line 62 “and cyclin B1 mRNA increased upon IGF1 treatment[21]”. The original article actually describes an increase in cyclin-D1 mRNA not in cyclin B1 mRNA. Maybe I’m missing something here.

3- Although the duration time for each sub-phase is shown in Figure 2B, it would be helpful if the authors could provide the mean values for the timing in prometaphase and in metaphase. Proper statistical analysis should be performed to compare control vs siRNA cells.

Line 147 replace “with” by “after”.

4- The 2h-treatment with IGF1R inhibitors further suggests that the delay in M-phase is mainly attributed to extended prophase/prometaphase. This was assessed by evaluating cell morphology following RO-3306. It would be important to monitor mitotic progression under these circumstances by live-cell microscopy.

5- Extended prometaphase duration suggests problems in kinetochore-microtubule attachments, whereas prolonged metaphase duration often results from permanent SAC signaling due to defects in SAC silencing mechanisms (please check PMID: 30446607 and PMID: 30736436). In that respect, can the authors investigate the accuracy of kinetochore-microtubule attachments in IGF1R-depleted or inhibited cells? I recommend calcium-treatment of prometaphase and metaphase cells to better visualize K-fibers.     

To obtain further insight into the cause of prolonged prometaphase and metaphase durations, it would be useful to evaluate the kinetochore levels of SAC proteins (Mad1 and BubR1 for example). Elevated levels of these proteins on metaphase kinetochores (that are attached to spindle microtubules) would indicate constitutive SAC activation despite proper attachments. This is symptomatic of impaired PP1 or PP2A activity, which could also explain the chromosome congression problems.

6- The authors examined the effects of combining IGF1R inhibition with the Aurora B inhibitor ZM447439 on cell proliferation. The number of cells was then evaluated by WST-8 assay and the authors concluded that SAC-targeting agents improve the suppressive effect of IGF1R inhibitors on cell proliferation. To strengthen this conclusion, I recommend directly assessing cell proliferation by plotting cell growth curves as Log10 of viable cell number over time.

To be able to generalize that inhibition of the SAC synergizes with the inhibition of IGF1R in suppressing cell proliferation, the authors should demonstrate that inhibition of the SAC with other drugs targeting other regulators (AZ3146 for Mps1 for example) produces a similar effect as when ZM447439 is used.    

7- Line 347 “In addition, prophase/prometaphase was prolonged in IGF1R knockdown cells, suggesting that IGF1R is involved in spindle formation and that IGF1R inhibition causes defects in spindle formation or function, activating the SAC.” There is no evidence in this manuscript to support that IGF1R inhibition causes defects in spindle formation or function. As mentioned in my point 5, the authors should evaluate whether loss of IGF1R is delaying the stabilization of kinetochore-microtubule attachments.

8- Please revise the English and grammar of the Discussion section.